# Is Oral Vaccination against *Escherichia coli* Influenced by Zinc Oxide?

**DOI:** 10.3390/ani13111754

**Published:** 2023-05-25

**Authors:** Guillermo Ramis, Francisco Murciano, Juan Orengo, Belén González-Guijarro, Amanda Cuellar-Flores, Daniel Serrano, Antonio Muñoz Luna, Pedro Sánchez-Uribe, Laura Martínez-Alarcón

**Affiliations:** 1Departamento de Producción Animal, Facultad de Veterinaria, Universidad de Murcia, 30100 Murcia, Spain; guiramis@um.es (G.R.); francisco.murcianor@gmail.com (F.M.); bgg@um.es (B.G.-G.);; 2Instituto Murciano de Investigación en Biomédicina (IMIB), 30120 Murcia, Spain; 3Departamento de Sanidad Animal, Facultad de Veterinaria, Universidad de Murcia, 30100 Murcia, Spain; 4ELANCO Animal Health, 28108 Alcobendas, Spain; 5Unidad para Docencia, Investigación y Calidad (UDICA), Hospital Clínico Universitario Virgen de la Arrixaca, 30120 Murcia, Spain

**Keywords:** *Escherichia coli*, oral vaccination, zinc oxide

## Abstract

**Simple Summary:**

Zinc oxide remains a widely used compound for the control and prevention of post-weaning *Escherichia coli* (*E. coli*) diarrhoea in piglets. It is sometimes administered concomitantly with oral *E. coli* vaccines. In this research, we assessed the influence of the administration of zinc oxide in piglet feed in combination with a bivalent vaccine against *E. coli*. We studied the immune activation, intestinal integrity, production of secretory IgA, and excretion of *E. coli* via faecal samples at different times post-vaccination. Although the main difference observed was the excretion of the *E. coli* vaccine strain, the immune response determined in both vaccine groups was similar, irrespective of the presence of ZnO in the feed.

**Abstract:**

Background: Although zinc oxide has been banned at therapeutic doses in the EU, its use is still legal in most countries with industrial pig farming. This compound has been shown to be very effective in preventing *E. coli*-related diseases. However, another strategy used to control this pathogen is vaccination, administered parenterally or orally. Oral vaccines contain live strains, with F4 and F18 binding factors. Since zinc oxide prevents *E. coli* adhesion, it is hypothesised that its presence at therapeutic doses (2500 ppm) may alter the immune response and the protection of intestinal integrity derived from the vaccination of animals. Methods: A group of piglets were orally vaccinated at weaning and divided into two subgroups; one group was fed a feed containing 2500 ppm zinc oxide (V + ZnO) for the first 15 days post-vaccination (dpv) and the other was not (V). Faeces were sampled from the animals at 6, 8, 11, 13, and 15 dpv. Unvaccinated animals without ZnO in their feed (Neg) were sampled simultaneously and, on day 15 post-vaccination, were also compared with a group of unvaccinated animals with ZnO in their feed (ZnO). Results: Differences were found in *E. coli* excretion, with less quantification in the V + ZnO group, and a significant increase in secretory IgA in the V group at 8 dpv, which later equalised with that of the V + ZnO group. There was also some difference in *IFNα*, *IFNγ*, *IL1α*, *ILβ*, and *TNFα* gene expression when comparing both vaccinated groups (*p* < 0.05). However, there was no difference in gene expression for the tight junction (TJ) proteins responsible for intestinal integrity. Conclusions: Although some differences in the excretion of the vaccine strain were found when comparing both vaccinated groups, there are no remarkable differences in immune stimulation or soluble IgA production when comparing animals orally vaccinated against *E. coli* in combination with the presence or absence of ZnO in their feed. We can conclude that the immune response produced is very similar in both groups.

## 1. Introduction

Piglets at weaning are confronted with numerous changes that often result in the transient but long-lasting activation of the immune system [1], inflammation, and destruction of the intestinal epithelium [2]. In addition to these physiological and immunological changes connatural to weaning itself, there is also the action of various pathogens present in the microbiota or in the environment that, taking advantage of these unfavourable conditions, can cause disease. One of the main actors in this field is *E. coli* [3,4], a multi-pathotype bacterium that produces up to five different pathological expressions: neonatal diarrhoea, colibacillary diarrhoea in young piglets, colibacillary post-weaning diarrhoea, oedemas disease, and septicaemic colibacillosis [5].

For years, the prevention of *E. coli* diseases has been based on the use of antimicrobials and zinc oxide [6]. However, these two treatments, although very efficient, can no longer be used due to public health concerns [7] and the generation of resistance in the former and environmental concerns in the latter. In fact, the use of zinc oxide at therapeutic levels was banned in the European Union (EU) in June 2022 [8]. Vaccination is another strategy that is currently being used, with both parenteral and oral vaccines available, both of which are effective [9,10,11] but produce different immune responses immediately after administration [12].

The use of zinc oxide at therapeutic doses is still legal in many parts of the world, at least during the first 15 days post-weaning. Vaccination against the pathogen is usually carried out at weaning, so it may be that oral vaccines are administered concurrently with the presence of zinc oxide at high doses (2500–3000 ppm) in piglets’ feed. Although zinc oxide is known to be very effective in the prevention of *E. coli*-related diseases, its mechanism of action has not been fully elucidated [13]; however, it is believed that high doses of zinc exceed the detoxification capacity of zinc by bacteria [14]. It is known that resistance to the action of this metal is not only different among species but may also be different between individuals of the same species [15]. In murine models, it has been shown that enterotoxigenic *E. coli* (ETEC) and enteroaggregative *E.coli* (EAEC) strains produce alterations in the adhesion and gene expression of virulence factors (including the production of mRNA for thermolabile (LT) toxin), as well as the activation of cytokine expression in the gut, thereby decreasing inflammation following infection by these pathogens [16,17,18]. In fact, human macrophages have been shown to use zinc-mediated toxicity to fight pathogens such as *Salmonella enterica* serovar *Typhimurium* or *Mycobacterium tuberculosis* [19]. Zinc ion bacteriolysis has been shown to be induced by activated peptidoglycans, amidases, endopeptidases, and carboxypeptidases [20]. However, zinc resistance mechanisms have also been demonstrated in human uropathogenic strains (UPEC) [15]. 

After oral vaccination with a bivalent F4/F18 vaccine, increased gene expression of cytokines such as IFNγ and TGFβ in the ileum, calprotectin in the jejunum and ileum, and occludin in the jejunum, ileum, and colon, as well as decreased mRNA for zonulin in the jejunum, ileum, and colon, have been observed [12].

Given this information, the bacteriolytic capacity of zinc oxide, its ability to prevent *E. coli*-related diseases, and even the possibility of certain strains showing resistance to zinc ions can be better understood. The aim of this work was to investigate the excretion of the vaccine strain, immune activation, and intestinal integrity of animals administered with a bivalent F4/F18 vaccine with and without zinc oxide at the therapeutic level in their feed.

## 2. Materials and Methods

### 2.1. Animals

One hundred sixty eight pure Large White piglets from the farm of the University of Murcia were used. The farm is in Guadalupe de Maciascoque (Murcia, Spain; 38.00662156425343 N, −1.1749983147290364 W). Piglets were subjected to usual farm management, including treatment with toltrazuril, iron dextran, and tooth-filing on the third day.

The animals were uniquely identified with numbered ear tags and were weaned at an age of 24 ± 1.2 days. After weaning, the animals were housed in nursery rooms of 6 pens with a capacity of 7 piglets per pen. One room per treatment was used, and each group included 42 animals housed in 6 pens. Feed was administered by means of a 4-hole hopper, with free disposal of feed, and water was supplied ad libitum by means of a nipple drinker.

Four groups were set up: a negative control group without vaccination or therapeutical levels of zinc oxide in their feed (Neg), a positive control non-vaccinated group with 2500 ppm zinc oxide (ZnO) in their feed, a vaccinated group without zinc oxide in their feed (V) and, finally, a vaccinated group with 2500 ppm zinc oxide in their feed (V + ZnO). Zinc oxide was administered during the first 15 days after weaning. A schematic of the experimental design is shown in Figure 1.

### 2.2. Vaccine and Vaccination

The animals were orally vaccinated with the Coliprotect F4 + F18 vaccine (Elanco GmbH, Monheim am Rhein, Germany), which contains live non-pathogenic *E. coli* strain O8:K87 and live non-pathogenic *E. coli* strain O141:K94 (1.3 × 10^8^ to 3.0 × 10^9^), thus ensuring the presence of antigens F4ac and F18ac. The animals were vaccinated the day after weaning, when the piglets had adapted to the nursery. The vaccine was reconstituted with mineral water and 2 mL was administered to each piglet individually by drenching.

The animals were housed in four rooms separated by stalls. The two feeds (with or without ZnO) used for the vaccinated groups were randomly assigned to each room.

### 2.3. Feeds

All feeds were produced using a basal diet, the formula for which is given in Table 1. To the feeds of the ZnO and V + ZnO groups, 2500 ppm of zinc oxide was added, while the basal diet had 150 ppm of zinc oxide. 

### 2.4. Weighing of Animals

Piglets were weighed at weaning, and at days 0, 8, 14, and 25 post-weaning, using a GRAM K2 scale (GRAM, Barcelona, Spain) suitable for weighing animals. It was decided not to weigh the animals every day of faecal sampling to avoid continuous stress to the animals. With these data, the average daily gain (ADG) was calculated over each of the periods (ADG_14_; from day 0 to 14 and ADG_25_: from day 0 to 25). Personal protective equipment (PPE) and specific boots were used to enter each of the rooms to avoid transferring bacteria between rooms.

### 2.5. Faecal Sampling

Faecal samples were taken from 20 animals from each group and divided into two portions: one 100 mg portion was preserved by RNAlater (Invitrogen, Waltham, MA, USA) for gene expression quantification and the other for DNA isolation for *E. coli* quantification. The former was kept refrigerated for 24 h after sampling and then kept at −80 °C until analysis; the portion used to obtain DNA was immediately placed in a −80 °C environment. The animals were left for 6 days following vaccination to allow for multiplication of the vaccine strain and, thereafter, they were sampled on 6, 8, 11, 13, and 15 days post-vaccination (dpv). Sampling ceased on day 15, as this was the last day the animals ate feed containing zinc oxide. The ZnO group was only sampled 15 dpv, as an indicator at the end of the study.

### 2.6. Isolation of Nucleic Acids

Nucleic acids were isolated by means of Genejet RNA purification and Genejet PCR purification kits (Thermo Fisher, Waltham, MA, USA). The quantity of DNA and RNA was quantified using a Nanodrop 2000 (Thermo Fisher, Waltham, MA, USA). Regarding the RNA samples, cDNA was synthesised using oligo dT as a primer in order to utilise only the mRNA in the synthesis.

### 2.7. Quantification of E. coli

For the quantification of E. coli, five qPCRs were used, including the EXOone *E. coli* internal reference for quantification of the total *E. coli* (ECOTotal), EXOone *E. coli* virulence factor F4, EXOone *E. coli* CRIPS, EXOone *E. coli* H7, and EXOone *E. coli* O8 qPCR kits (Exopol, Zaragoza, Spain), according to the supplier’s instructions. A Quantum Studio 5 thermocycler (Life Techologies, Carlsbad, CA, USA) was used. The PCR profile was 95 °C 10 min, 95 °C 1 min, 60 °C 30″, and 42 cycles. 

### 2.8. Gene Expression for Intestinal Integrity and Immune Stimulation

Relative quantification of the gene expression of *calprotectin*, *zonulin,*
*occludin*, *claudin*, *IL1α*, *IL1β*, *IL6*, *IL8*, *IL10*, *IL12p35*, *IL12p40*, *IFNα*, *IFNγ*, *TGFβ*, and *TNFα* was performed using the methodology and primers previously described [12,21,22], and using *β-actin* as a housekeeping gene (Table 2). All relative quantifications were carried out on faecal samples. 

### 2.9. Secretory IgA Quantification

The secretory IgA content of the RNAlater used to preserve the faeces was quantified. The preservative was recovered before RNA isolation, centrifuged at 3900× *g* for 10 min, and preserved at −80 °C until analysis. IgA quantification was performed as described in the literature [12,31] using the Pig IgA ELISA kit (Bethyl Laboratories, Montgomery, TX, USA) according to the manufacturer’s specifications.

### 2.10. Statistical Analysis

All data were analysed using the SPSS program (SPSS Inc., Chicago, IL, USA). The results of the production data were analysed by means of a one-way ANOVA. For the comparison of quantifications, a Kruskal–Wallis test was used, as well as a subsequent Mann–Whitney U test to constitute a two-samples-comparison analysis. Violin plots were produced for comparison of results with GraphPad Prism v. 8.2 (GraphPad, San Diego, CA, USA). In addition, correlations among parameters were calculated as partial correlations controlled for group and sampling. The significance level and tendency were set at *p* ≤ 0.05 and *p* ≤ 0.10, respectively.

For data reduction, Discriminant Function Analysis (DFA), which uses Wilks’ lambda to determine the efficiency of analysis and generates as many functions as it deems necessary to explain the model, was used. A dot plot was generated when two or more functions were produced.

## 3. Results

### 3.1. Weights and Growth

The mean weights found for each group in each sampling are shown in Figure 2.

Average daily gains from day 0 to 14 (ADG_14_) and from day 0 to 25 (ADG_25_) are shown in the table below (Table 3).

A significant difference in ADG_14_ was found when comparing the Neg group with all other groups, while there was no difference between the ZnO and V groups, and there was a significant difference between the ZnO and V + ZnO groups and between the V and V + ZnO groups (Table 2). For ADG_25_, the main differences were between the ZnO groups with the V group and the V + ZnO group and between the V and V + ZnO groups. In this case, the highest ADG corresponded to the ZnO group, followed by Neg and the two vaccinated groups, which obtained the lowest ADG_25_. In the first part of the experiment, the vaccinated groups demonstrated significantly greater growth than the two control groups, but in the second half of the study, there was compensatory growth, especially in the ZnO group. This shows that even in the absence of vaccination or *E. coli* prophylaxis, animals eventually adapt and recover homeostasis and compensatory growth.

### 3.2. Quantification of E. coli

#### 3.2.1. Frequency of Positive Samples

The frequencies of positive samples for each *E. coli*-related gene tested, sorted by sampling day, are shown in the table below (Table 4).

As expected, none of the Neg samples showed amplification for CRISPR, as this gene is only present in the vaccine strain. This indicates that there was no transfer from the vaccine strain to the Neg group throughout the experiment. On the other hand, none of the samples from the Neg group were positive for adhesion factor F4. These two findings suggest that all samples with F4 amplification were amplified as a consequence of vaccination, since this fimbria is present in the vaccine strain. The main differences were found in the 6 and 8 dpv samples, for which group V showed higher frequencies for F4 and CRISPR. On day 8 post-vaccination, both vaccinated groups had an increased frequency of F4, the V + ZnO group had a slight increase in CRISPR frequency, and both vaccinated groups had an increased frequency of O8, while the Neg group had reduced frequencies of positive samples. On 11, 13, and 15 dpv, there were differences in the frequency of O8-positive samples when comparing the vaccinated groups and the V + ZnO group with the Neg group. In the vaccinated groups, the frequency of this gene decreased throughout the experimental period while, in the Neg group, it increased. 

The frequency of CRISPR-positive samples showed a bimodal distribution over the duration of the experiment for the V group, but not for the V + ZnO group; in the V group, 55% of the pigs excreted *E. coli* with this gene in the first sampling. This percentage was maintained on day 8, decreased to 15% on day 11, and increased again to 35% on day 13 before decreasing slightly on day 15. In the V + ZnO animals, the frequency of CRISPR-positive samples started from 40% 6 dpv, increased to 55% 8 dpv, dropped to 15% on day 11, but did not increase as in the V group, instead remaining at 15% until the end of the experimental period (Figure 3).

The Ct for ECOTotal for each group and for each sampling date is shown in Figure 4.

#### 3.2.2. Quantification of *E. coli* Genes

The quantification of *E. coli* (ECOTotal) showed significant differences between the vaccinated and control groups at all sampling dates except 11 dpv, with a higher Ct in the V + ZnO group. The difference between the Neg and V + ZnO groups increased linearly (r = 0.924, *p* = 0.025) from 3.7 Ct (6 dpv) to 8.8 Ct (15 dpv). In the case of the comparisons between Neg and V groups, there were also differences in Ct except for the 6 dpv and 11 dpv samples, ranging from 0.2 cycles (6 dpv) to 4.84 more cycles at 15 dpv; this decrease was also linear, although not as marked or significant as in the comparison with the V + ZnO group (r = 0.794, *p* = 0.108) (Figure 5).

After analysing the H7 gene, a significant difference was observed 15 dpv when comparing the Neg group with the V group (*p* = 0.001). However, when comparing the Neg group with the V + ZnO group, there were differences at 11 dpv (*p* = 0.002), 13 dpv (*p* = 0.033), and 15 dpv (*p* = 0.002). Differences between the V and V + ZnO groups were observed on days 11 (*p* < 0.001) and 13 (*p* = 0.016).

The CRISPR gene was not amplified in any of the samples from the Neg group, as shown above. Comparing the V and V + ZnO groups, no significant differences were found, although, in all cases, the Ct was higher in the V + ZnO group. Comparing the Ct, there was a linearly increasing difference (r = 0.924, *p* = 0.076) between 6 dpv (1) and 13 dpv (6.33). Finally, on day 15, this difference was limited to one cycle.

There were no differences between the Ct obtained for the Neg, V, and V + ZnO groups for the O8 and F4 genes in any of the analyses. None of the samples analysed from the Neg group were positive for F4.

When conducting a paired samples analysis, there were differences between the Ct for ECOTotal in group V when comparing 6 dpv quantification with that of 13 dpv (*p* < 0.001) and with that of 15 dpv (*p* < 0.001), as well as when comparing 11 dpv with 13 dpv samples (*p* = 0.01). In the V + ZnO group, however, differences were found when comparing quantification at 6 dpv with those at 11 dpv (*p* < 0.001), 13 dpv (*p* = 0.001), and 15 dpv (*p* = 0.011), and when comparing Ct at 8 dpv with those at days 11 (*p* < 0.0001), 13 (*p* = 0.001), and 15 (*p* = 0.001). The Neg group had no difference in Ct when comparing any of the samples.

When analysing the Ct for the H7 gene for each sampling date, differences were observed in group V when comparing the 8 dpv sample with the 15 dpv sample (*p* = 0.006), and when comparing the 11 dpv sample with 13 dpv (*p* = 0.003) and 15 dpv (*p* = 0.010) samples. In the V + ZnO group, differences were found when comparing the Ct at 6 dpv with those at 11 dpv (*p* = 0.016), 13 dpv (*p* = 0.004), and 15 dpv (*p* = 0.05), and when comparing the 8 dpv sample with the 13 dpv sample (*p* = 0.001). Regarding CRISPR gene quantification, differences were only found in the V + ZnO group when quantification at 6 dpv was compared with that at 8 dpv (*p* = 0.008).

No differences were found between samples in any of the experimental groups for F4 and O8. 

Positive correlations were found between all genes used to quantify *E. coli* excretion (Table 5).

Regarding total *E. coli* quantification, the highest correlation was for the H7 sample, with a similar correlation for F4 and CRISPR, again suggesting that F4 is acquired from vaccination, supported by the absence of amplification in the Neg group samples (Table 4).

### 3.3. Intestinal Integrity

Quantifications for the three TJ proteins are shown in Figure 6.

For occludin, a fourfold increase in gene expression was found for the V group when compared to the Neg group (*p* = 0.016) at 6 dpv. At 8 dpv, a 19- and 30-fold increase in gene expression was observed in the vaccinated groups (*p* < 0.001), a difference that was not maintained at 11 dpv but was again 24- and 30-fold higher at 13 dpv (*p* = 0.001). At 15 dpv, there was a 9-fold and 25-fold reduction in the vaccinated groups compared to the ZnO group (*p* = 0.027), but differences were found when compared to the Neg group or in comparisons of the Neg group with the ZnO group.

A similar pattern was observed for zonulin, with significant 330- and 500-fold increases in the vaccinated groups at 6 dpv (*p* < 0.001), 16,980- and 21,671-fold increases at 8 dpv (*p* < 0.001), 623- and 500-fold increases at 11 dpv (*p* < 0.001), and 850- and 1300-fold increases at 13 dpv (*p* < 0.001). However, at 15 dpv, the vaccinated groups had a 1.55- and 1.3-fold decrease compared to the Neg group (*p* = 0.033), but there was no difference between the vaccinated groups and the ZnO group.

Finally, for claudin, there was a 6-fold reduction when comparing V and Neg groups (*p* = 0.001) at 6 dpv, a 5.2-fold reduction at 8 dpv (*p* = 0.044), a 33-fold reduction at 11 dpv (*p* = 0.05), and an 82-fold reduction at 13 dpv (*p* < 0.001), a well as a 2.3-fold increase in the V group compared to the V + ZnO group at 6 dpv (*p* = 0.013), a 2.33-fold increase at 8 dpv (*p* = 0.017), a 7.4-fold increase at 11 dpv (*p* = 0.014), and a 25-fold increase at 13 dpv (*p* < 0.001). At 15 dpv, all groups had a decrease in gene expression compared to the ZnO group: 69-fold for the Neg group (*p* = 0.016), 172-fold for the V group (*p* < 0.001), and 62-fold for the V + ZO group (*p* < 0.001).

### 3.4. Immune Stimulation

The changes in gene expression by groups and sampling dates are shown in Figure 7.

Regarding calprotectin, there were only differences in the vaccinated groups versus the Neg group with higher quantification in the two vaccinated groups, and with an increase of 52- and 4-fold more than the Neg group (*p* = 0.004 and *p* < 0.001). At 15 dpv, the V group had a trend towards significance with 32 times more mRNA than the ZnO group (*p* = 0.062). At 6 dpv, the V + ZnO group had a 2.33-fold increase in quantification over the V group (*p* = 0.046).

IFNα had differences between the three groups at 6 dpv, with higher quantification in the Neg group and a 20-fold and 5-fold difference with V and V + ZnO groups (*p* < 0.001). At 8 dpv, there was a 48-fold increase in the V + ZnO group over the V group (*p* < 0.001) and at 13 dpv, the V group had26- and 1024-fold differences with the V + ZnO and Neg groups (*p* = 0.002). At 11 dpv, there were no differences between groups and at 15 dpv, there were differences between all groups with the highest quantification in the Neg and ZnO groups and the lowest in the two vaccinated groups with differences of 300- and 32-fold less in the V group (*p* < 0.001) and 91- and 9-fold less in the V + ZnO group (*p* < 0.001).

For *IFNγ*, significant differences in quantification were found in the two vaccinated groups compared to the Neg group at 6 dpv with 64-fold increases (*p* < 0.001), at 8 dpv with more than 4000-fold increases (*p* < 0.001), at 11dpv with a 16-fold and 8-fold increase, respectively (*p* < 0.001), and at 13 dpv with a 256-fold increase in mRNA quantification (*p* < 0.001). At 15 dpv, the V + ZnO group showed a 5-fold increase over the Neg group (*p* = 0.022) and among the vaccinated groups with a 5-fold increase in the V group compared to the V + ZnO group (*p* = 0.030). No difference was observed when comparing the ZnO group with all others. At 11 dpv, a 3-fold increase was also determined in the V group compared to the V + ZnO group (*p* = 0.046).

For *IL1α*, a 9-fold increase was found in the two vaccinated groups compared to the Neg group (*p* < 0.001) at 8 dpv and a 12- and 8-fold increase at 13 dpv (*p* < 0.001 and *p* = 0.002, respectively). At 15 dpv, the highest quantification was observed in the ZnO group with a 13-fold increase over the Neg group (*p* = 0.01) and 4500- and 14000-fold increases compared to V and V + ZnO groups (*p* < 0.001), and there was a trend towards a difference between the V and Neg groups with 1.8-fold more mRNA in the vaccinated groups (*p* = 0.065). With respect to *IL1β*, there were differences between the vaccinated and Neg groups at 6 dpv with a 21.5- and 7-fold reduction (*p* < 0.001), at 13 dpv with a 20- and 8-fold reduction (*p* < 0.001), and at 15 dpv with a 315- and 69-fold reduction with respect to the Neg group (*p* < 0.001). Differences were also found between the V and V + ZnO groups at 8 dpv with a 5-fold increase for the V + ZnO group compared with the V group. In all samplings, the V group had the lowest quantification and the Neg group had the highest with the V + ZnO group’s quantification falling between them.

*IL6* showed lower quantification in the vaccinated groups compared to the Neg group at 11 dpv (19- and 8.7-fold decrease; *p* = 0.001) and at 15 dpv (2280- and 67-fold decrease; *p* < 0.001), but higher quantification at 8 dpv (13- and 27-fold increase over Neg; *p* < 0.001). At 15 dpv, both vaccinated groups showed a lower quantification than the ZnO group (3.7- and 2.8-fold increase; *p* < 0.001).

*IL8* gene expression only showed a fourfold decrease in the vaccinated groups compared to the Neg group at 11 dpv (*p* = 0.028). *IL10* showed an 8.44- and 2.6-fold reduction in V and V + ZnO groups compared to the Neg group at 6 dpv (*p* = 0.004 and *p* = 0.049, respectively) and a 34- and 67-fold reduction at 13 dpv (*p* = 0.049) with no further significant differences observed.

*IL12p35* had a 30-fold lower quantification in the V + ZnO group compared to the Neg group (*p* = 0.015) at 6 dpv and both vaccinated groups showed an 88- and 42-fold reduction over the Neg group at 11 dpv (*p* < 0.001). However, both vaccinated groups had 29- and 31-fold higher gene expression than Neg at 8 dpv (*p* < 0.001) and 1.4- and 7-fold higher gene expression at 13 dpv (*p* < 0.001). At 15 dpv, the V group showed a 32-fold reduction over the V + ZnO group (*p* = 0.002) and a 19-fold reduction over the Neg group (*p* = 0.002). *IL12p40* showed no difference at 6 dpv, but there was a 52- and 29-fold increase in the V and V + ZnO groups compared to the Neg group (*p* < 0.001) at 8 dpv and an 8- and 15-fold increase at 13 dpv (*p* = 0.005). However, at 11 dpv, the two vaccinated groups showed a 5- and 4-fold decrease compared to the Neg group (*p* = 0.004). Finally, at 15 dpv, mRNA quantification was 3.1, 2, and 5 times higher in the ZnO group than in the V, V + ZnO, and Neg groups (*p* = 0.001).

*TGFβ* expression increased 121- and 153-fold in the V and V + ZnO groups compared to the Neg group (*p* < 0.001) at 8 dpv and 30- and 33-fold at 13 dpv (*p* < 0.001), while there were no differences in the other samplings.

Finally, a 4- and 2.6-fold decrease in *TNFα* expression was observed in the vaccinated groups compared to the control (*p* < 0.001) at 6 dpv, and a 23- and 24-fold increase at 8 dpv (*p* < 0.001). The V + ZnO group had a 1.5- and 4-fold increase compared to Neg and ZnO groups at 13 dpv (*p* < 0.001). At 15 dpv, there was a 78- and 50-fold reduction in the vaccinated groups compared to the Neg group (*p* < 0.001) and a 5.5- and 3.7-fold reduction compared to the ZnO group (*p* < 0.001).

### 3.5. Secretory IgA Quantification

The quantifications of secretory IgA in samples from groups V and V + ZnO are shown in Figure 8. Significant differences between groups were found only at 8 dpv (*p* = 0.047). There was a much more rapid increase in the amount of secretory IgA in the V group than in the V + ZnO group. In fact, in the V group, there is a 159% increase in the amount of secretory IgA, while in the V + ZnO group, in the same period, there is a 7% increase in the amount of antibody. Between 8 dpv and 11 dpv, the V + ZnO group experiences an increase of 108% in the amount of IgA. The increase in secretory IgA in the V + ZnO group was slower, and this produces the difference observed at 8 dpv.

### 3.6. Correlations among Parameters

#### 3.6.1. Correlations among *E. coli* Gene Quantification and Performances

There was no correlation among performances and *E. coli* gene quantification. There were also no differences in the different weights recorded when comparing animals testing positive or negative for any of the *E. coli* genes studied (*p* > 0.05).

#### 3.6.2. Correlations among *E. coli* Gene Quantification and Immune Stimulation Gene Expression

The correlations between the quantifications for cytokine gene expression and Ct for *E. coli* genes are shown in Table 6.

#### 3.6.3. Correlations among *E. coli* Genes and Intestinal Integrity

The correlations between the quantifications for TJ protein gene expression and Ct for *E. coli* genes are shown in Table 7. Only a partial correlation controlled for group and sampling date was found between the Ct obtained for O8 and the relative mRNA quantification for zonulin (Table 7).

### 3.7. Data Reduction including Immune System and Intestinal Integrity Biomarkers

The Wilks’ lambda showed significance for all samples (*p* < 0.001) and two or more functions were produced, except for the sampling at 15 dpv, where only one function was produced and explained 100% of the variance (*p* = 0.003) (Figure 9). 

The presentation of the DFA data showed that the Neg group was always far away from the vaccinated groups while, between the vaccinated groups, there was no significant distance after applying the functions. This reinforces the idea that vaccination in the presence or absence of ZnO produces a clear effect compared to non-vaccinated animals.

The group assignment is shown in the following table (Table 8).

## 4. Discussion

The influence of the presence of zinc oxide in feed, both in native high-dose and microencapsulated low-dose forms, has been extensively demonstrated both scientifically and empirically in field trials [32,33,34]. However, in this study, we found significantly higher growth during the first 14 days post-weaning in the groups vaccinated with or without zinc oxide in their feed. However, thereafter, compensatory growth occurred and the overall growth at 25 days was higher in the ZnO group than in the vaccinated group alone. It should be noted that the vaccinated animals did not present any health problems related to *E. coli* and no sample containing *E. coli* F4 was found in the control group. Therefore, in this case, it may be that the vaccine gave an initial advantage immediately after weaning, but at around 25 days post-weaning, the energy consumption due to the immune response, in the absence of pathogen-related problems, slightly detracts from growth efficiency. Some cases have been documented where the addition of ZnO to feed did not result in better growth compared to untreated groups [33] or even groups vaccinated orally against *E. coli* or with a symbiotic in their feed [35]. It has also been observed in different experiments with oral vaccination against *E. coli* F4 + F18 that there were no differences in ADG when comparing vaccinated animals with unvaccinated and untreated control animals [10]. Thus, Vangroenweghe’s study recorded ADG values very similar to those observed in the current research. Compensatory growth comparing the first two weeks of rearing with the following two weeks has already been described with other feed additives such as symbiotics, similar to the effect observed in this study in the ZnO group [35]. In other cases, the lower ADG observed in vaccinated vs. control piglets, under commercial conditions, has been attributed to a reduction in antibiotic consumption in vaccinated vs. unvaccinated control animals [36], but no antibiotics were used in either group in our study, so we can rule out this type of effect.

Despite its almost universal use for the prevention of *E. coli*-related diarrhoea, it has also been shown that the addition of zinc oxide at 2500 ppm has a moderate effect on F4 strains [37] and there is no difference in the presence of faecal *E. coli* compared to animals without zinc oxide in their feed [38]. However, it has also been shown that in the presence of zinc oxide in feed, there is a reduction in the excretion of *E. coli* and a reduction in the frequency of diarrhoea [39]. 

Studies show that the presence of ZnO in *E. coli* F4 infections reduces the expression of pro-inflammatory cytokines such as *IFNγ, TNFα*, and *IL6* [32,34] and increases *TGFβ* gene expression [40]. It has been shown that such activation of pro-inflammatory cytokines has a negative effect on the proteins of the immune system, especially *IFNγ, TNFα,* and *IL1β* [1,41,42,43]. However, we should note that the animals have been vaccinated in this study, which produced variations in the gene expression of cytokines, key elements for the activation of the immune system. The presence of calprotectin mRNA in faeces following oral *E. coli* vaccination has been demonstrated previously [35,44] and results from the recruitment of antigen-presenting cells such as macrophages and neutrophils necessary to produce the immune response to the vaccine. It has previously been determined that oral vaccination with the vaccine used in this study resulted in increased *IFNγ* and *TGFβ* gene expression in the ileum and faeces, detectable at 6, 8, 14, and even 21 dpv in tissues compared to unvaccinated animals [12,35,44]. In the current study, vaccinated groups showed higher gene expression for both biomarkers than Neg or ZnO groups, significant at 8 and 13 dpv. Interestingly, *TNFα* is not above that of the Neg group until 8 dpv, and again at 13 dpv, while at 15 dpv, both vaccinated group values are below those of the Neg and ZnO groups. In the case of *IFNγ*, there was more mRNA in all quantifications except at 15 dpv, which may be related to the immune activation produced by vaccination. The fact that, at 15 dpv, both vaccinated group values were lower than those of the control groups may be the result of the increase in *TGFβ* observed at 13 dpv.

Interestingly, many of the immune system biomarkers showed a bimodal response, with an initial increase comparing 6 and 8 dpv, a subsequent decrease at 11 dpv, a further increase at 13 dpv, and a final decrease. This type of response is also observed in the frequency of CRISPR-positive samples, at least in the V group and in the Ct for ECOTotal. The cyclic increase in *E. coli*, including the vaccine strain, could be responsible for this bimodal response. 

However, zinc does not only exert actions on the immune system or intestinal integrity or innate immunity. It is also capable of preventing bacteria from adhering to the intestinal epithelium, which could be one of the pillars of zinc’s action against *E. coli*. This non-adhesion may be key to preserving the integrity of the TJ [40]. We have seen how TJ proteins were increased in the two vaccinated groups compared to the Neg group at 6, 8, and 11 dpv in the case of occludin, and on days 6, 8, 11, and 13 dpv in the case of zonulin. This would indicate that vaccination is producing a preservation of intestinal integrity even though immune activation is occurring. The effect of oral vaccination with the same vaccine on zonulin and occludin at 6 and 8 dpv, with increased gene expression compared to unvaccinated animals, has been previously described [44]. 

The efficacy of vaccination with bivalent F4 + F18 vaccines can lead to competition and the displacement of wild-type strains, as vaccine strains bind to recipients, thereby displacing wild-type strains and producing an IgM- and IgA-based immune response [4,9]. Precisely, the presence of zinc oxide could reduce the number of bacteria that bind to the receptors and, thus, could reduce the multiplication of bacteria and the effect on the immune system. When comparing the quantification of the studied *E. coli* genes, a linear decrease in the Ct for ECOTotal was observed in the two vaccinated groups, in contrast to the Neg group. However, this decrease was much more pronounced in the V + ZnO group, with a difference of more than 8 cycles compared to the Neg group, which meant almost 2.7 exponentials less. However, the V group had, at most, 4.8 cycles of difference at 15 dpv when compared with the Neg group, which was 1.5 exponentials less; the reduction had a much steeper slope than that observed for the V + ZnO group. When comparing the paired samples for CRISPR, a significant difference was observed when comparing the first quantification on day 6 with the second quantification at 8 dpv in the V + ZnO group with a difference of 3.7 cycles (more than one exponential), while the comparison in the V group showed a difference of only 1 cycle, meaning that half the number of bacteria were excreted. All these data indicated that the V + ZnO group excreted less *E. coli* in all quantifications and that there was a significant linear decrease in the amount of faecal *E. coli* excreted. In group V, the same effect occurred but was less significant. Our findings of decreased *E. coli* excretion following vaccination are consistent with those previously reported by Nadeau et al. [9]. However, a challenge was made in that work with a pathogenic strain and not in ours.

Our team previously determined the difference in the amount of secretory IgA in orally vaccinated animals vs. unvaccinated control animals or even parenterally vaccinated 21 dpv animals [12], findings consistent with other research [4,9]. However, while Nadeau et al. [9] found an increase in IgA around day 21, in this study, we observed an increase as early as day 8, although this increase remained at the same level until day 15 and was lower than that documented by us previously in similar animals on the same farm. It should be noted that, in the previous study [12], secretory IgA was studied in tissue samples, in contrast to the current study, which investigated secretory IgA in faeces, as it is difficult to compare the levels observed in tissue samples and faeces. 

## 5. Conclusions

Although zinc oxide is, empirically, very effective in the control of *E. coli*-related diarrhoea, the results obtained in this study do not allow us to affirm that a clearly different response is produced after administering a bivalent oral vaccine in the presence or absence of zinc oxide in feed fed to the piglets immediately after vaccination. Differences were observed in the excretion of *E. coli*, including the vaccine strain, especially at the end of the study period, as well as differences in the presence of secretory IgA in the second sampling and in the gene expression of *TNFα*, *IL1β*, *IL1α*, *IFNγ*, or *IFN α*, but only in some of the samples and not consistently.

Although the discriminant capacity of FDA was high, there was a wide percentage of samples that the functions could not correctly assign to a vaccine group. We can conclude that there were no relevant differences in immune activation, IgA production, or intestinal integrity between groups of vaccinated animals depending on whether they took zinc oxide in their feed or not. It must be emphasised that this study was not an efficacy study, there was no experimental infection, and the animals did not suffer any pathological events related to *E. coli.* In fact, the Neg group did not have any positive samples for F4, one of the adhesion factors present in the vaccine and, therefore, there was no effect derived from the prevention of the action of these strains.

## Figures and Tables

**Figure 1 animals-13-01754-f001:**
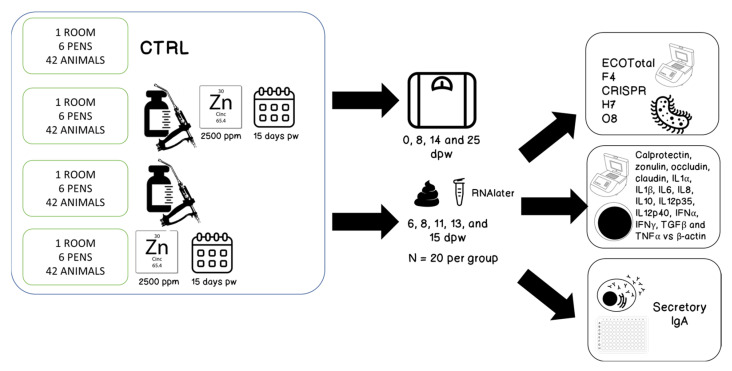
Schematic representation of the experimental design. Pw = post-weaning, dpw = days post-weaning.

**Figure 2 animals-13-01754-f002:**
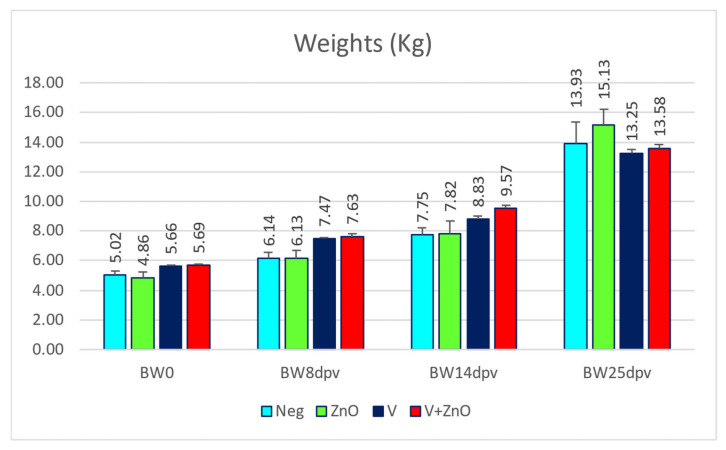
Body weight (BW) at weaning (WB0), 8, 14, and 25 dpv sorted by groups. The bars represent mean ± SEM.

**Figure 3 animals-13-01754-f003:**
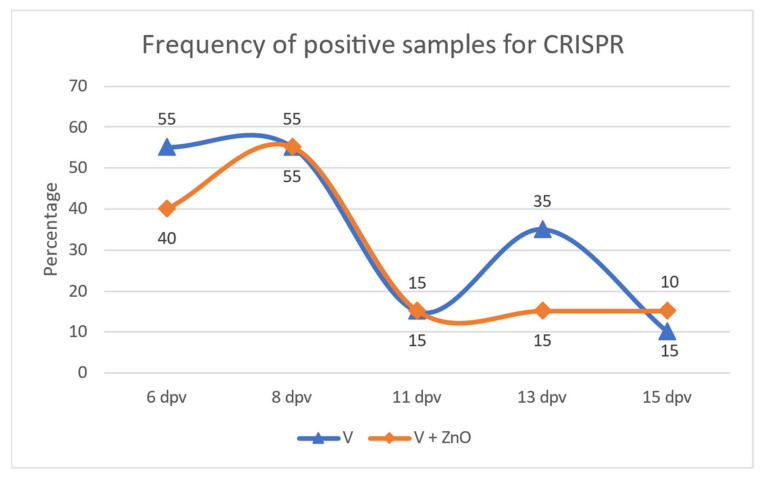
Frequency of positive samples for CRISPR over the experimental period.

**Figure 4 animals-13-01754-f004:**
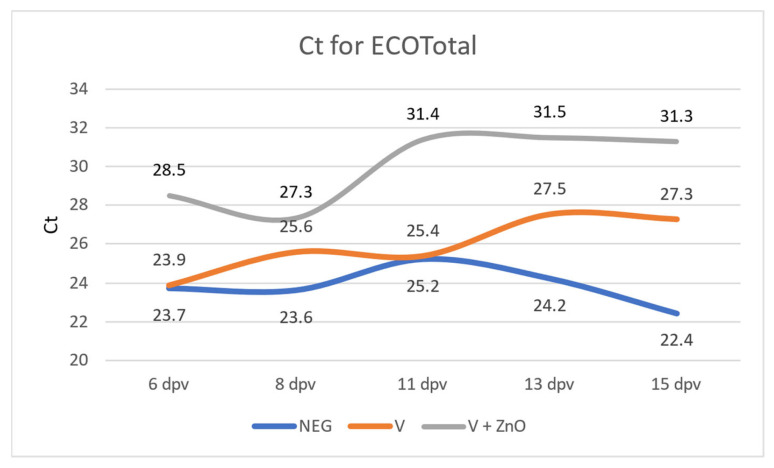
Ct for ECOTotal for every group and sampling date.

**Figure 5 animals-13-01754-f005:**
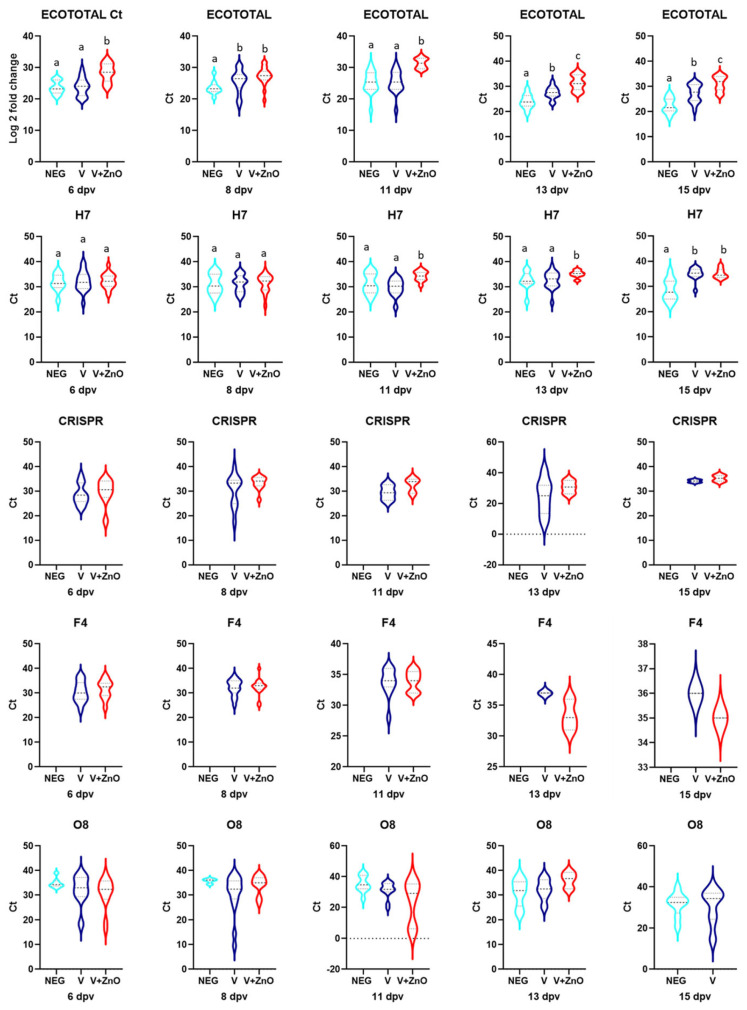
Violin plots for Ct calculated for each experimental group in each sampling. Different superscripts mean significant difference (*p* < 0.05).

**Figure 6 animals-13-01754-f006:**
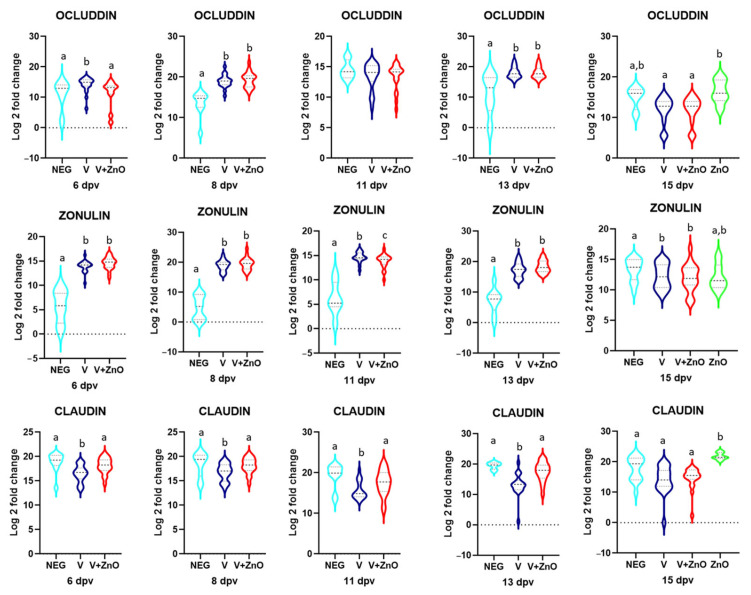
Violin plots for Log 2 of the relative quantification for occludin, zonulin, and claudin mRNA on each sampling day. Different superscripts above each violin within each sampling day indicate significant differences between groups (*p* < 0.05).

**Figure 7 animals-13-01754-f007:**
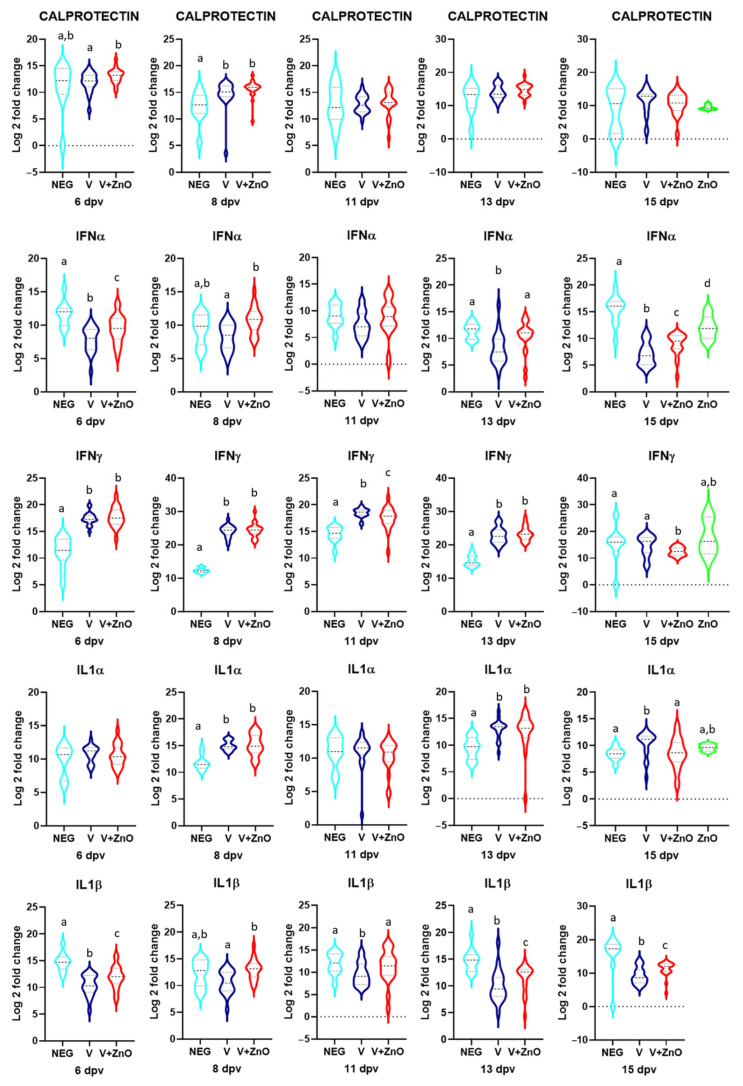
Violin plots for Log 2 of the relative quantification *for Calprotectin*, *IFNα*, *IFNγ*, *IL1α*, *IL1β*, *IL6*, *IL8*, *IL10*, *IL12p35*, *IL12p40*, *TGFβ*, and *TNFα* mRNA on each sampling day. Different superscripts above each violin within each sampling day indicate significant differences between groups (*p* < 0.05).

**Figure 8 animals-13-01754-f008:**
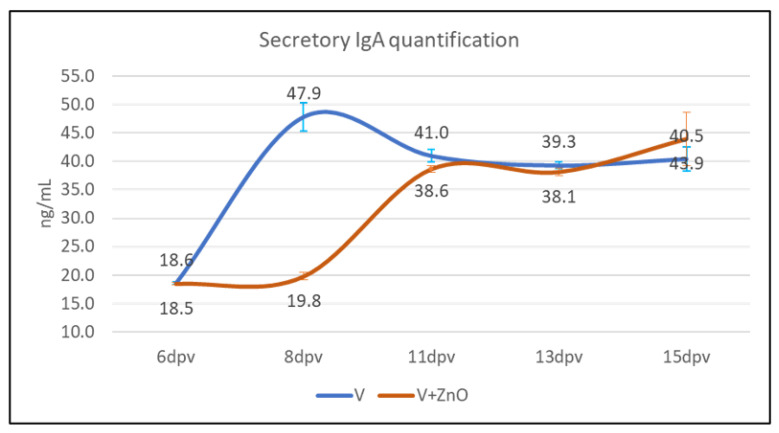
Secretory IgA content in faeces preservative for V and V + ZnO groups over the experimental period. Dots represent mean ± SEM.

**Figure 9 animals-13-01754-f009:**
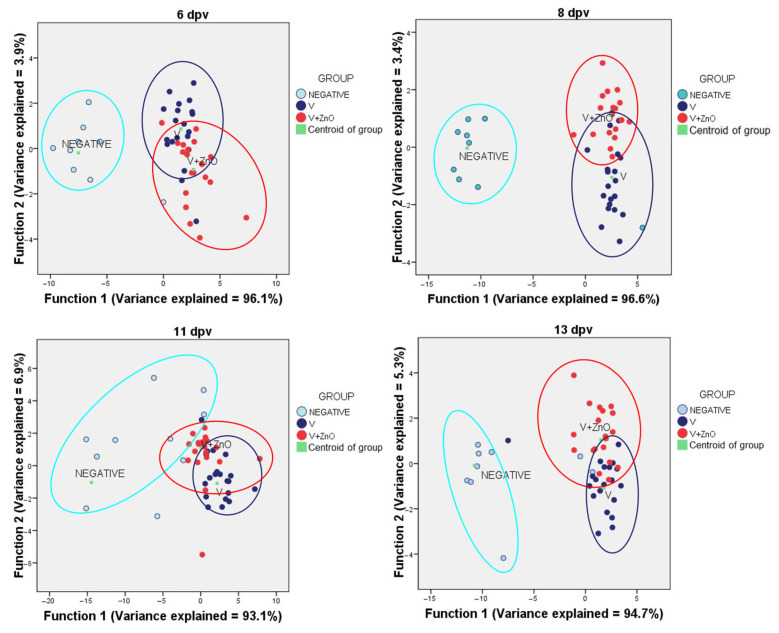
Dot plot for the DFA using all the immune system and intestinal integrity biomarkers.

**Table 1 animals-13-01754-t001:** Basal diet formula used in all experiments.

Rawstaff	Percentage
Corn 13%	25.00
Wheat F10	25.00
Malting barley 9.5	15.72
Soymeal 46%	10.05
Processed soy protein HP300	6.00
Porcine protein hydrolysate (Palbio 50 RD)	3.75
Sheep sweet whey	3.60
Refatted whey 50	2.50
Whey bran	2.50
Soy oil F10	2.36
L-Lysine 50	0.95
Calcium carbonate	0.69
Monocalcium phosphate	0.62
Vitamin-mineral corrector ^1^	0.30
L-Threonine	0.26
DL-Methionine	0.24
Salt	0.22
L-Valine	0.10
L-Tryptophan	0.08
Mycotoxin absorber	0.05

^1^ Supplied per kg of complete diet: vitamin A (3a672a), 12,000 IU; vitamin D3 (3a671), 2000 IU; vitamin E (3a700), 80 mg; vitamin K3 (3a710), 2 mg; vitamin B1 (3a821), 1.5 mg; riboflavin, 4 mg; vitamin B6 (3a831), 2.5 mg; vitamin B12, 0.025 mg; niacin (3a315), 25 mg; folic acid (3a316), 0.5 mg; biotin (3a880), 0.1 mg; choline chloride (3a890), 220 mg; pantothenic acid (3a841), 13 mg; vitamin C (3a300), 100 mg; Iron (II) chelate of the amino acid glycine (3b108), 7.7 mg; Mn (3b503), 40 mg; Zn (3b603), 120 mg; Fe (3b101), 120 mg; Cu (3b405), 150 mg; I (3b202), 0.65 mg; Se (E8), 0.25 mg; L-Valine (3c371), 1700 mg; L-Tryptophan (3c440), 800 mg; Endo-1,4-β-xylanase (EC 3.2.1.8), 200 FXU; 6-phytase (EC 3.1.3.26), 500 FYT.

**Table 2 animals-13-01754-t002:** Primers for *Occludin*, *zonulin 1*, *claudin 1*, *calprotectin*, *IL-1β*, *IL-6*, *IL-8*, *IL-10*, *IL-12p35*, *IL-12p40*, *TNF-α*, *IFN-α*, *IFN-γ*, and *TGF-β* and *β-Actin* (housekeeper gene).

Gene	Primer Forward (5′ → 3′)	Primer Reverse (5′ → 3′)	References
*Occludin*	5′-TTGCTGTGAAAACTCGAAGC-3′	5′-CCACTCTCTCCGCATAGTCC-3′	[12]
*Zonulin 1*	5′-CACAGATGCCACAGATGACAG-3′	5′-AGTGATAGCGAACCATGTGC-3′	[12]
*Claudin 1*	5′-ACCCCAGTCAATGCCAGATA-3′	5′-GGCGAAGGTTTTGGATAGG-3′	[21]
*Calprotectin*	5′-AATTACCACGCCATCTACGC-3′	5′-TGATGTCCAGCTCTTTGAACC-3′	[12]
*IFN-* *α*	5′-CCCCTGTGCCTGGGAGAT-3′	5′-AGGTTTCTGGAGGAAGAGAAGGA-3′	[23]
*IFN-* *γ*	5-TGGTAGCTCTGGGAAACTGAATG-3′	5′-GGCTTTGCGCTGGATCTG-3′	[24]
*TNF-* *α*	5′-ACTCGGAACCTCATGGACAG-3′	5′-AGGGGTGAGTCAGTGTGACC-3′	[25]
*IL-12p35*	5′-AGTTCCAGGCCATGAATGCA-3′	5′-TGGCACAGTCTCACTGTTGA-3′	[23]
*IL-12p40*	5′-TTTCAGACCCGACGAACTCT-3′	5′-CATTGGGGTACCAGTCCAAC-3′	[26]
*IL-10*	5′-TGAGAACAGCTGCATCCACTTC-3	5′-TCTGGTCCTTCGTTTGAAAGAAA-3′	[24]
*TGF-* *β*	5′-CACGTGGAGCTATACCAGAA-3′	5′-TCCGGTGACATCAAAGGACA-3′	[23]
*IL-8*	5′-GCTCTCTGTGAGGCTGCAGTTC-3′	5′-AAGGTGTGGAATGCGTATTTATGC-3′	[27]
*IL-1* *α*	5′-GTGCTCAAAACGAAGACGAACC-3′	5′-CATATTGCCATGCTTTTCCCAGAA-3′	[28]
*IL-1* *β*	5′-AACGTGCAGTCTATGGAGT-3′	5′-GAACACCACTTCTCTCTTCA-3′	[29]
*IL-6*	5′-CTGGCAGAAAACAACCTGAACC-3′	5′-TGATTCTCATCAAGCAGGTCTCC-3′	[29]
*β* *-actin*	5′-CTACGTCGCCCTGGACTTC-3′	5′-GATGCCGCAGGATTCCAT-3′	[30]

**Table 3 animals-13-01754-t003:** Average daily gain (ADG) at the 14th and 25th days of experiment, sorted by groups.

Parameter	Neg	ZnO	V	V + ZnO	*p*-Value
ADG_14_ (Kgd^−1^)	0.180 ± 0.006 ^a^	0.217 ± 0.013 ^b^	0.227 ± 0.006 ^b^	0.277 ± 0.007 ^c^	<0.001
ADG_25_ (Kgd^−1^)	0.347 ± 0.054 ^a,b^	0.388 ± 0.040 ^a^	0.304 ± 0.007 ^b^	0.288 ± 0.014 ^c^	0.045

^a,b,c,^ Different superscripts in the same row indicate significant differences (*p* < 0.005). Data represent mean ± SEM.

**Table 4 animals-13-01754-t004:** Frequency of ECOTotal-, F4-, H7-, CRISPR-, and O8-positive samples. The *p*-value indicates differences between expected and observed frequencies and AR = adjusted residual for positive samples. The *p*-values indicating statistically significant differences are marked in bold.

Sampling dpv	Group	Neg	V	V + ZnO	*p*-Value
6	ECOTotal	100% (10/10)	100% (20/20)	90% (18/20)	NS
	F4	0% (0/10) AR = −3.7	70% (14/20) AR = 2.1	60% (12/20)	**<0.001**
	H7	100% (10/10)	90% (18/20)	85% (17/20)	NS
	CRISPR	0% (0/10) AR = −2.8	55% (11/20) AR = 2.0	40% (8/20)	**0.003**
	O8	70% (7/10)	85% (17/20)	70% (14/20)	NS
8	ECOTotal	100% (10/10)	100% (20/20)	100% (20/20)	NS
	F4	0% (0/10) AR = −3.7	85% (17/20) AR = 2.1	85% (17/20) AR = 2.1	**<0.001**
	H7	100% (10/10)	100% (20/20)	85% (17/20) AR = −2.2	**0.055**
	CRISPR	0% (0/10) AR = −3.1	55% (11/20)	75% (15/20)	**0.001**
	O8	40% (4/10) AR = −2.7	55% (11/20)	90% (18/20) AR = 2.1	**0.013**
11	ECOTotal	100% (10/10)	80% (16/20)	75% (15/20)	NS
	F4	0% (0/10)	55% (11/20)	45% (9/20)	**0.013**
	H7	100% (10/10)	65% (13/20)	55% (11/20)	NS
	CRISPR	0% (0/19)	15% (3/20)	15% (3/20)	NS
	O8	30% (4/10)	65% (13/20) AR = 2.2	30% (6/20)	**0.027**
13	ECOTotal	100% (10/10)	100% (20/20)	90% (18/20)	NS
	F4	0% (0/10)	5% (1/20)	15% (3/20)	NS
	H7	90% (9/10)	90% (18/20)	70% (14/20)	NS
	CRISPR	0% (0/10)	35% (7/10) AR = 2.2	15% (3/20)	**0.027**
	O8	100% (10/10) AR = 4.3	30% (6/20)	15% (3/20) AR = −2.7	**<0.001**
15	ECOTotal	100% (10/10)	100% (20/20)	65% (17/20)	**0.055**
	F4	0% (0/10)	5% (1/20)	5% (1/20)	NS
	H7	100% (10/10) AR = 2.9	60% (12/20)	40% (8/20) AR = −2.4	0.007
	CRISPR	0% (0/10)	10% (2/20)	10% (2/20)	NS
	O8	80% (8/10) AR = 2.7	65% (13/20) AR = 2.7%	0% (0/20) AR = −4.9	**<0.001**

**Table 5 animals-13-01754-t005:** Partial correlations controlled for experimental group and sampling day between the observed Ct for the different genes.

	F4	H7	CRISPR	O8
ECOTotal	** *0.502* **	** *0.717* **	** *0.531* **	0.231
F4		** *0.809* **	** *0.549* **	** *0.574* **
H7			** *0.576* **	** *0.515* **
CRISPR				0.325

Correlations marked in bold and italics were significant at the 0.01 level.

**Table 6 animals-13-01754-t006:** Partial correlations controlled for experimental group and sampling day among the observed Ct for the different *E. coli* genes and the gene expression for the immune stimulation biomarkers.

	ECOTotal	F4	H7	CRISPR	O8
CALP	0.068	0.210	0.315	−0.081	0.306
IFNα	−0.305	−0.110	−0.202	**0.589**	−0.049
IFNγ	−0.018	0.063	0.278	−0.067	0.345
IL1α	−0.189	0.007	−0.003	−0.101	0.289
IL1β	−0.280	−0.184	−0.225	**0.592**	−0.092
IL6	0.237	0.092	0.332	−0.039	0.092
IL8	−0.262	0.000	−0.080	−0.164	0.244
IL10	−0.229	−0.195	−0.277	−0.433	−0.177
IL12p35	−0.062	0.062	0.202	−0.105	0.202
IL12p40	−0.039	0.071	0.229	−0.134	0.208
TGFβ	−0.118	0.008	0.133	−0.115	0.319
TNFα	0.035	0.019	0.196	−0.144	0.140

A positive correlation was found between the quantification of IFNα (*p* = 0.044) and IL1β (*p* = 0.043) with the observed CRISPR Ct. The correlations marked in bold were significant at 0.05 level.

**Table 7 animals-13-01754-t007:** Partial correlations controlled for experimental group and sampling day among the observed Ct for the different *E. coli* genes and the gene expression for the TJ protein biomarkers.

	ECOTotal	F4	H7	CRISPR	O8
OCLD	−0.174	−0.115	0.022	−0.158	0.265
ZON	−0.259	0.069	0.084	−0.142	**0.587**
CLAU	−0.241	−0.294	−0.103	0.374	−0.129

The correlations marked in bold were significant at 0.05 level.

**Table 8 animals-13-01754-t008:** Percentage of group membership assertion using DFA-calculated functions.

Day	Original	Predicted Membership Group (%)		
		Neg	V	V + ZnO
6	Neg	70.0	10.0	20.0
	V	10.0	75.0	15.0
	V + ZnO	5.0	40.0	55.0
8	Neg	90.0	10.0	0.0
	V	0.0	80.0	20.0
	V + ZnO	0.0	10.0	90.0
11	Neg	40.0	0.0	60.0
	V	0.0	80.0	20.0
	V + ZnO	0.0	20.0	80.0
13	Neg	80.0	10.0	10.0
	V	5.0	85.0	10.0
	V + ZnO	0.0	15.8	84.2
15	Neg	100.0	0.0	0.0
	V	0.0	90.0	10.0
	V + ZnO	0.0	31.6	68.4

The cases correctly grouped according to the original assignment were 66% at 6 dpv, 86% at 8 dpv, 72% at 11 dpv, 83.7% at 13 dpv, and 86.13% at 15 dpv.

## Data Availability

Data are not available.

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
