# Peer review of "Is Oral Vaccination against *Escherichia coli* Influenced by Zinc Oxide?"

_animals, 2023, doi:10.3390/ani13111754_

Round 1

Reviewer 1 Report

General comments

This is a well written manuscript that evaluate the effect of a modified-live oral vaccination against Escherichia coli with or without the addition of Zinc Oxide for 15 days post weaning. Weight, average daily gain, bacterial fecal shedding, immune stimulation and mucosal intestinal integrity gene expression and IgA in feces were evaluated in four groups of animals (negative control, vaccinated, vaccinated + Zinc Oxide in feed, and Only Zinc Oxide in feed). The rationale of manuscript is stated in the introduction and the objectives are clear. There are some specific aspects that were not so clear in the material and methods, which will need clarification. The results are demonstrated in the text and illustrated in the figures and tables. There are some aspects that need for clarification that will be mentioned below. Overall, the manuscript brings new information regarding the association of oral vaccination against E. coli and Zinc Oxide in feed.

Specific comments:

Line 20: Delete “...excretion…” after E. coli.

Lines 40-41: Conclusions are too vague and confusing. Rewrite.

Line 69: “… among …” instead of “… between…”

Lines 96-100: How many pens and animals were used per group? It is critical to state this information.

Lines 134-135: Is the period ADG25 from day 0 to 25 or 15 to 25? It is important to clarify this point.

Lines 160-162: It is important to state what will be the clinical sample used to detect gene expression.

Figure 1: The mean weight of pigs from group Neg at the BW25pv is difficult to see to the standard deviation line.

Line 241: “…observed on 15 dpv when…”

Figura 4: Graphic of H7, 15 dvp is stated 13 dvp instead.

Lines 291-293: Delete the following sentence “Finally, for claudin, there were…. vaccinated animals.” as it is incorrect.

Lines 325-326: Delete the sentence “At 15 dpv, …. groups (p=0.066).” It is repeating what was stated in the previous sentence.

Lines 339-340: It is not possible to understand which comparison of groups are mentioned. Rewrite the sentence.

Lines 341-342: “IL12p40 was similar at 8 and 13 dpv but was significantly higher…”

Line 343: “… groups at 8 and 13 dpv, and …”

Lines 345-346: Did increased for V+ZnO compared to the other groups at which time points?

Line 361: “…Figure 7. Significant…”

Line 363: Substitute “… more progressive..” by “… slower…”

Lines 369, 370, 372, 373, 376, 382, 384: E. coli in italic.

Line 384: “… table 6.” Not table below!

Three references are incomplete, missing the name of the journal: lines 557, 563, 618.

It will require some editing

Author Response

Thank you very much for your comments and feedback on our article. They have certainly helped us to improve it considerably.
Please find attached a document with detailed answers to the specific questions raised by the reviewer.
Kind regards
Guillermo Ramis
Juan Orengo 

Reviewer 2 Report

The aim of this study was to research the immunological influence of zinc oxide on the outcome of an oral E. coli vaccination for piglets. As the article is current, I recommend "major revisions" (extensive editing) to be considered for publication. Please find my comments below:

Line 17- Line 20: Please divide this sentence in two. The first sentence will establish the objective. The second sentence establishes what you have researched.

Line 21- 22 – Please reword “there was no strong evidence of relevant differences”. Reword what you mean because the sentence is too ambiguous.

Line 22- Line 23 – This sentence is too ambiguous. Rewrite the specific result/ conclusion in a simple way.

Line 35 – 41 – Please define in the results section of the abstract if the results were statistically significant or not. Please specify P>0.05 or P<0.05 after each statement. Do this rather than stating ambiguous sentences or using “there were some differences”.

Line 39- Please put the tight junctions’ abbreviations in parentheses.

Line 40 – Please rephrase the conclusion. Either there is evidence or not or make a more specific statement linking your main finding. As this sentence is written it is not a conclusion statement.

Line 59 – Write what the EU abbreviations stand for then write (EU) in parenthesis.

Line 76- Use the correct term for Salmonella enterica serovar Typhimurium. Fix the extra point after Mycobacterium.

Line 78- 79 – Do not start the sentence with the word “but”.

Line 86 – Make the aim one independent sentence.

Line 101-105- Please explain the experimental setup in a table.

Line 152- 162 -For qPCR, there are no specified primers. Make a table with the name of the primers, the sequence of the primers, the pair’s annealing temperature that was used for this experiment, and a reference if the primer was taken from previous literature.

Line 169- Specify the housekeeping gene that was used and include it in the table.

Line 185 – It is not specified if the error bars represent the +SEM. Change this to represent the ±SEM and include this in the graph description as well as a statistical statement, e.g., Different superscripts indicate significant differences (P<0.05). This is an important statistical remark even though there were no statistically significant results for this parameter.

Line 195- 197 – The sentence is a broad statement. Explain why it is interesting.

Line 278- 298- Explain the results in terms of fold change differences between group and control or between groups. Re-write this section.

Line 301- Line 303 – There is no statistical statement in the figure description. Please fix this.

Line 304 - Explain the results in terms of fold change differences between group and control or between groups. Re-write this section.

Line 359- 364 – Specify the increasing percentage in the results.

Line 366-367- The figure legend is incomplete. Add a statistical statement in the description.

Line 369 and 370- Italicize E. coli.

Line 370- 372 – Specify the p-value when stating there were no significant differences. For e.g., “there were no significant differences……. (p>0.05)”.

Line 373-376- Italicize E. coli.

Figure 8 – Please increase the font and image quality.

Other comments:

I recommend the authors proofread the entire manuscript as there are grammatical errors. The English grammar should be carefully revised by a native English speaker or grammar software. The authors use too many broad sentences and ambiguous wording throughout the entire manuscript.  

The result section in the manuscript should be improved as well as some figure legends. 

I recommend the authors proofread the entire manuscript as there are grammatical errors. The English grammar should be carefully revised by a native English speaker or grammar software. 

Author Response

(The authors gave the same response as above.)
